# Multifunctional Superparamagnetic Stiff Nanoreservoirs for Blood Brain Barrier Applications

**DOI:** 10.3390/nano9030449

**Published:** 2019-03-17

**Authors:** Zulema Vargas-Osorio, Andrés Da Silva-Candal, Yolanda Piñeiro, Ramón Iglesias-Rey, Tomas Sobrino, Francisco Campos, José Castillo, José Rivas

**Affiliations:** 1NANOMAG Laboratory, Applied Physics Department, Universidade de Santiago de Compostela, 15782 Santiago de Compostela, Spain; voz_wolverine74@hotmail.com (Z.V.-O.); jose.rivas@usc.es (J.R.); 2Clinical Neurosciences Research Laboratory, Health Research Institute of Santiago de Compostela (IDIS), Hospital Clínico Universitario, Universidade de Santiago de Compostela, 15706 Santiago de Compostela, Spain; Ramon.Iglesias.Rey@sergas.es (R.I.-R.); Tomas.Sobrino.Moreiras@sergas.es (T.S.); Francisco.Campos.Perez@sergas.es (F.C.); Jose.Castillo.Sanchez@sergas.es (J.C.)

**Keywords:** theranostic nanostructure, MRI contrast, neurological diseases

## Abstract

Neurological diseases (Alzheimer’s disease, Parkinson’s disease, and stroke) are becoming a major concern for health systems in developed countries due to the increment of ageing in the population, and many resources are devoted to the development of new therapies and contrast agents for selective imaging. However, the strong isolation of the brain by the brain blood barrier (BBB) prevents not only the crossing of pathogens, but also a large set of beneficial drugs. Therefore, an alternative strategy is arising based on the anchoring to vascular endothelial cells of nanoplatforms working as delivery reservoirs. In this work, novel injectable mesoporous nanorods, wrapped by a fluorescent magnetic nanoparticles envelope, are proposed as biocompatible reservoirs with an extremely high loading capacity, surface versatility, and optimal morphology for enhanced grafting to vessels during their diffusive flow. Wet chemistry techniques allow for the development of mesoporous silica nanostructures with tailored properties, such as a fluorescent response suitable for optical studies, superparamagnetic behavior for magnetic resonance imaging MRI contrast, and large range ordered porosity for controlled delivery. In this work, fluorescent magnetic mesoporous nanorods were physicochemical characterized and tested in preliminary biological in vitro and in vivo experiments, showing a transversal relaxivitiy of 324.68 mM^−1^ s^−1^, intense fluorescence, large specific surface area (300 m^2^ g^−1^), and biocompatibility for endothelial cells’ uptake up to 100 µg (in a 80% confluent 1.9 cm^2^ culture well), with no liver and kidney disability. These magnetic fluorescent nanostructures allow for multimodal MRI/optical imaging, the allocation of therapeutic moieties, and targeting of tissues with specific damage.

## 1. Introduction

Improvements in imaging techniques, or new therapeutic paradigms, are behind the continuous evolution of medicine towards the objective of defeating diseases. Nanomedicine has become, in recent years, a field of growing demand and the use of nanotechnology allows an approach at the molecular level, the same length scale as biological events (nanometres), with a strong focus [1] on developing biomedical agents with less secondary effects than traditional pharmaceuticals.

Designed nanoparticles offer the possibility of incorporating within a single carrier complementary biomedical functions [2,3], like therapy, imaging, and tagging, which allows for detailed studies at precise locations or specific treatments that avoid systemic dosage.

However, neurological diseases pose a great challenge in the field [4], in both the medical and scientific context, due to the intricate complexity of the central nervous system (CNS). The brain is a fragile organ composed of a dense network of billions of nerve cells, strongly protected by the blood brain barrier (BBB). Composed of tightly connected vascular endothelial cells, by adherents and junctions (TJs), and a sparse layer of pericytes, the BBB strictly regulates brain homeostasis [5]. This physiologic behavior is crucial for our survival, since mainly all pathogens are filtered out, but it is also the main hindrance for neuroscientific developments because it also filters out most drugs after systemic administration [6,7].

To facilitate the delivery and permeation of drugs to the brain, an emergent paradigm seeks to bind theranostic agents to the luminal plasma membrane of the BBB endothelial cells, thus using them as docking reservoirs. In this way, the binding to the endothelium allows nanoparticles to mark vessels affected by diverse cerebral pathologies with a vascular component and the development of targeting strategies to receptors that mediate transcytosis to allow the crossing of nanostructures through the BBB.

Although soft nanostructures (liposomes, thermoreversible polymers) have been preferred in the past as nanodelivery systems, their scarce colloidal stability and immune system response preclude them from being effective for clinical therapies. Therefore, there is a need of multifunctional nanocarriers with high biocompatibility and stability under forced flow or diffusion through highly viscous tissues for different in vivo applications. In this regard, mesoporous nanostructures combine a set of properties that ensure their suitable performance for different theranostic purposes, with the advantage of being chemically stable and resistant under variable pH, temperature, and flow conditions.

Among their advantages, the following have been reported [8,9,10]: Low immune response, chemical resistance to enzymes [11]; prolonged drug release [12,13,14,15]; large surface area for gene transfection [16,17], bone regeneration [18,19], and multimode theranostic actions [20,21,22]; magnetic separators [23]; cell markers [24]; and supports for enzyme immobilization [25].

Moreover, enhanced biological advantages can be achieved by tailoring the intrinsic characteristics of mesoporous silica nanostructures [26], such as porosity, functionalization [27,28], size [29], and shape. Specifically, it has been reported that rod-like shaped mesoporous silica nanoparticles are well suited for monitoring cell trafficking [30], and their hydrophilic character facilitates the adhesion and functioning of several brain cell types [31].

In this work, the development, physicochemical study, and in vitro and in vivo tests of mesoporous silica nanorods are presented. Rod-like nanostructures with large pores (d = 7 nm) obtained by a soft template method were functionalized with amino groups and a fluorescent dye and coated with superparamagnetic magnetite NPs. Their physicochemical characterization shows a set of multifunctional abilities, like good colloidal stability, transversal relaxivitiy of r^2^ = 324.68 mM^−1^ s^−1^, intense fluorescence emission, and large porosity (300 m^2^ g^−1^). Biological characterizations show good biocompatibility for endothelial cell uptakes up to 100 µg/culture plate for in vitro tests and no brain, liver, and kidney disability after in vivo administration.

This preliminary physicochemical and biological in vitro/in vivo study shows that magnetic mesoporous silica nanorods present a combination of properties that make them suitable candidates to be tested as multifunctional nanoreservoirs in brain endothelium applications.

## 2. Materials and Methods

### 2.1. Chemicals and Synthesis Chemicals

3-Aminopropyl(diethoxy)methylsilane (≥97%), tetraethyl orthosilicate (98%), iron(III) chloride hexahydrate (97%), hydrochloric acid (37%), phosphoric acid (85%), cyclohexane (99.8%), Igepal CO-520 [Polyoxyethylene (5) nonylphenylether, branched], glycerol solution (86–89%), Pluronic^®^ P123[triblock-copolymer-PEO_20_:PPO_70_:PEO_20_,Poly(ethyleneglycol)-*block*-poly(propyleneglycol)-*block*-poly(ethyleneglycol)], average M_n_ ≈ 5800), Rhodamine B isothiocyanate (C29H30CIN3O3S, RBITC), isooctane (C_8_H_18_ ≥ 99%), 2-propanol (≥99.5%, IPA), Tween^®^ 20 (viscous liquid, polyethylene glycol sorbitan monolaurate), paraformaldehyde (PFA, reagent grade, crystalline), and potassium hexacyanoferrate(II) trihydrate (ACS reagent, 98.5–102.0%) were obtained from Sigma Aldrich (3050 Spruce Street, Saint Louis, MO, USA); iron(II) sulfate heptahydrate (99%) and ammonium hydroxide (28%) from Fluka Analytical (Honeywell Fluka™, Morris Plains, NJ, USA); oleic acid (extra pure) from Merck (Darmstadt, Germany); ethanol (99.9%) and acetone (≥99%) were purchased at Scharlau (Senmanat, Spain). Phosphate buffer solution (PBS, pH 7.4), Dulbecco’s Modified Eagle Medium (DMEM), fetal bovine serum, and penicillin-streptomicin from Gibco-Invitrogen, (Gibco-Invitrogen™-Fisher, Carlsbad, CA, USA). All commercial chemicals did not require any further purification. All water utilized in the experiments was Milli-Q (Millipore^®^, Burlington, MA, USA) deionized water.

### 2.2. Synthesis of Fluorescent Magnetic Mesoporous Nanorods

Rod-like SBA-15 mesoporous silica (S15N nanorods) were prepared following the method used by Wang and coworkers [32]. Triblock copolymer, Pluronic P123, was used as a template in the presence of glycerol solution, while tetraethyl orthosilicate (TEOS) was the silicate precursor in acidic conditions. In a typical synthesis, 7.8 g of both P123 and glycerol were dissolved in 300 g of an aqueous acidic solution 2 M of HCl/H_3_PO_4_ (2:1). The mixture was stirred at 35 °C until the surfactant was completely dissolved. Then, 16.4 mL TEOS (80.61 mmol) was added to the above solution under vigorous magnetic stirring. After 10 min, the stirring was stopped, and the reaction continued in static conditions for 24 h, followed by aging at 100 °C for 24 h. The final product was filtrated, washed with water, and dried at 60 °C. The resulting material was extracted with a mix of isooctane/ethanol and water/acetone to remove the surfactant. Magnetite nanoparticles (NPs) were obtained by the coprecipitation method, in which 90 mmol of FeCl_3_·6H_2_O and 60 mmol of FeSO_4_·7H_2_O were dissolved in 200 mL of 0.01 M HCl aqueous solution and mechanically stirred. The mixture was heated up to 60 °C, then 1500 mmol of NH_4_OH and 14.2 mmol of oleic acid were added, and the reaction was carried out for 1 h. After that, the sample was transferred to a beaker and placed on a hot plate at 100 °C to allow for precipitation. The precipitate containing oleic acid-capped magnetite nanoparticles [Fe_3_O_4_@OA], OAM, was retained with a magnet and the supernatant was removed. OAM were washed three times with deionized water.

Afterwards, magnetic S15N nanorods, HMMSN, were obtained by covalently anchoring magnetite NPs over mesoporous silica nanorods [33]. For this, a previously prepared solution (2.4 mL) of oleic acid-capped magnetite nanoparticles [Fe_3_O_4_@OA] stabilized in cyclohexane at 3.5% (*w*/*v*) was poured into a 7.6% solution of Igepal CO-520 in cyclohexane (266.16 mmol), at a 1:12.5 volume ratio, and stirred for 30 min. Then, 12.56 mmol of NH_4_OH and 0.5 g of S15N nanorods were added to the mixture and stirred for 40 min until the amino functional coupling agent (1.35 mmol) was dispensed dropwise into the mixture. After keeping the reaction isolated from the light at room temperature for 16 h, reactants were precipitated with isopropyl alcohol (50 mL), and magnetically separated. Subsequent washing steps were needed to clean the sample, four with ethanol and five with deionized H_2_O. Finally, the resulting solid was filtered and dried at 60 °C for 24 h.

Linking of fluorescent RITC to HMMSN was performed by adding under magnetic stirring a solution (V = 200 µL) of RBITC to ethanol (20 mL), with concentration (2.5 mg/mL), and kept in a flask in an ice bath under N_2_ atmosphere for 30 min. Afterwards, 100 mg of HMMSN dispersed in 15 mL of ethanol was dropped into the RBITC solution, and kept for 24 h under N_2_ atmosphere, glycerol reflux at 2 °C, and protected from light. After the reaction was completed, the fluorescent material was magnetically separated and washed first with a mixture of IPA/acetone and finally with distilled water by sonication and centrifugation. The sample was dried at 60 °C for 5 h and then by a desiccator.

### 2.3. Physicochemical Characterization

#### 2.3.1. XRD-Structural Characterization

Crystalline phases of the HMMSN hybrid magnetic mesoporous nanorods (HMMSN) were characterized by X-ray powder diffraction (XRD) using a Philips PW1710 diffractometer (Cu Kα radiation source, λ = 1.54186 Å) with 2θ measurements between 10–80° and scanning steps of 0.02° and 10 s/step, on powdered samples. Low-angle X-ray diffraction (XRD) patterns of the inner channels’ structure of the mesoporous silica nanorods were recorded on a PANalytical X’Pert Powder Empyrean, which was used to analyze the rodlike mesoporous SBA-15 silica structure, using a 2θ in the range of 0.25–6°, and a step size of 0.01° (5 s/step).

#### 2.3.2. Microscopy Morphological Characterization

Transmission electron microscopy (TEM) images were collected on a JEOL JEM-1011 microscope (Peabody, Peabody, MA, USA, 100 kV). Sample morphology was characterized by scanning electron microscopy (SEM), using a Zeiss FE-SEM ULTRA Plus microscope (Carl Zeiss AG, Oberkochen, Germany) operated at 5 kV. The optical properties of the nanorods were measured using a fluorescent spectrophotometer (Varian Cary Eclipse, Agilent, Santa Clara, CA, USA).

Fluorescent nanoparticles imaging was performed with a fluorescence confocal multispectral imaging confocal laser microscope, Leica TCS SP8 SMD. TECHNICAL SPECIFICATIONS: Resolution: 1024 × 1024; scan direction X: Bidirectionnal; objetive: HC PL APO CS 63 × /1.40 OIL; and laser line: 552 nm

Cells micrographs were imaged in a Leica DMI 6000 B microscope with the software, LAS AF 1.0.0 (Leica Microsystems, Wetzlar, Germany).

#### 2.3.3. Surface Chemistry Characterization

Fourier transform infrared (FTIR) spectra of the surface functional groups of the nanorods were recorded with a Thermo Nicolet Nexus spectrometer (Nicolet, Hirsemarken, Denmark) using the attenuated total reflectance (ATR) method. The silicon chemical environments were determined by ^29^Si solid-state magic-angle spinning nuclear magnetic resonance (MAS NMR). The NMR spectra were obtained on a Bruker Advance 400 with a spectral width of 32 kHz. A 4.0-mm rotor was used. The MAS speed was 5 kHz. ^29^Si MAS NMR spectra were referenced to tetramethyl silane (TMS).

#### 2.3.4. Compositional Characterization

The composition of the samples was analyzed by elemental microanalysis (CHNS) in a LECO CHNS-932 thermoanalyzer (St. Joseph, MI, USA).

#### 2.3.5. Textural Characterization

Nitrogen sorption experiments were performed at 77 K using a Quantachrome Autosorb IQ2 (Boynton Beach, FL, USA) and the specific surface areas were estimated from the Brunauer-Emmett-Teller (BET) method. The pore size distributions were calculated from the desorption branch of the isotherms by means of the Barrett-Joyner-Halenda (BJH) method.

#### 2.3.6. Magnetic Characterization

Magnetization was measured on dried HMMSN samples, with a vibrating sample magnetometer (VSM) at room temperature and applied magnetic fields from −10 to +10 kOe.

Magnetic resonance imaging was performed with a 9.4 T horizontal bore magnet (Bruker BioSpin, Ettlingen, Germany) with 440 mT/m gradients and a combination of a linear birdcage resonator (7 cm in diameter) for signal transmission and a 2 × 2 surface coil array for signal detection. A quadrature volume coil (7 cm in diameter) was also used in phantom studies. MRI post-processing was performed using ImageJ software (W. Rasband, NIH, Bethesda, MD, USA).

Relaxivity of the HMMSN samples was measured in Agar phantoms loaded with D-HMMSN nanorods, which were prepared following a previous protocol with different Fe concentrations: 0.2, 0.1, 0.05, 0.02, 0.01, and 0.005 mM. T2-weighted images were acquired using a multi-slice multi-spin-echo sequence (MSME) with a 10.44 ms echo time, 3 s repetition time, 16 echoes with 10.4 ms echo spacing, 50 KHz spectral bandwidth, flip angle (FA) = 110°, 14 slices of 1 mm, 1 average, field of view (FOV) of 75 × 75 mm^2^ (with saturation bands to suppress the signal outside this FOV), a matrix size of 256 × 256 (in-plane resolution of 293 μm pixel^−1^ × 293 μm pixel^−1^), and implemented without the fat suppression option. T2*-weighted images were acquired using a multi gradient echo (MGE) sequence with a 4.44 ms echo time, 1.8 s repetition time, 16 echoes with 6.75 ms echo spacing, 100 KHz spectral bandwidth, FA = 30°, 14 slices of 1 mm, 2 averages, and the same geometry parameters as that of the T2-weighted images.

### 2.4. Biological Characterization

#### 2.4.1. In Vivo Magnetic Resonance Imaging

In vivo imaging of the HMMSN nanorods after the intraparenquimal injection was performed obtaining T2-weighted images by using an RARE-VTR sequence with the following acquisition parameters: Echo time = 9.5 ms, 8 echos, rare factor = 2, repetition time = 3 s, number of averages = 2, field-of view = 19.2 × 19.2 mm^2^, image matrix = 192 × 192 (isotropic in-plane resolution of 100 μm/pixel × 100 μm/pixel), and 18 consecutive slices of a 0.5 mm thickness. T2*-weighted images were acquired using an MGE sequence with 8 echos, first echo time = 3.13 ms, echo spacing = 3.38 ms, repetition time = 1.4 s, number of averages = 2, and the same geometry parameters as that of the T2-weighted images. In order to evaluate possible ischemic lesions in the head of the animals after intravenous and intra-arterial injection, T2-weighted images were acquired using an MSME sequence with a 9 ms echo time, 3 s repetition time, 16 echoes with 9 ms echo spacing, FA = 180°, 2 averages, 75 KHz spectral bandwidth, 14 slices of 1 mm, 19.2 × 19.2 mm^2^ FOV (with saturation bands to suppress the signal outside this FOV), a matrix size of 192 × 192 (isotropic in-plane resolution of 100 μm/pixel × 100 μm/pixel), and implemented without the fat suppression option.

#### 2.4.2. In Vitro Cell Mortality

Microvasculature brain endothelial cells (ATCC CRL-2299 Bend.3 cells) in passages between 2 and 5 were used to evaluate the in vitro toxicity of the HMMSN nanorods. Cells were cultured in DMEM with 10% (*v*/*v*) of fetal bovine serum and 0,1% (*v*/*v*) of penicillin-streptomicin in a 24 wells plate with a 1.9 cm^2^ cell growing area (Corning, Corning, NY, USA), and incubated at 37 °C under a humidified atmosphere containing 95% air and 5% of CO_2_. Media were replaced every 3 days until 80% of confluence was reached. A total of four different amounts of nanoparticles were added to each plate well (N = 8 well/group), 200, 100, 50, and 25 μg, and incubated for 24 h; then, the culture was washed 3 times with PBS pH 7.4 to remove non-bound/-uptaken particles. Cells were cultured with normal media for an additional period of 12 h. Once concluded, 200 μL of media per well were collected to perform an LDH toxicity assay. (Thermo Fisher Scientific ref# 88953). A positive control for cell death was used to determine the 100% of mortality and LDH release by adding the lysis buffer provided by the kit, while another group with only cells was used as a control for normal growth and viability.

#### 2.4.3. Prussian Blue Stain and Imaging

Cells previously incubated with the HMMSN nanorods were gently washed with PBS, permeabilized for 1 h with PBS and 0.01% (*v*/*v*) Tween 20, and fixed with 4% (*v*/*v*) of methanol free PFA. Later, the culture was washed and covered with Prussian Blue working solution composed by an aqueous solutions mixture (1:1) of HCl at 20% (*w*/*v*) and potassium hexacyanoferrate (II) trihydrate at 10% (*w*/*v*). After 30 min, cells were washed with PBS and imaged. Microphotographs at 10×, 20×, and 40× were randomly taken from samples using a phase contrast filter.

#### 2.4.4. In Vivo Experimental Groups

All animal studies were conducted with Male Sprague-Dawley rats of 325–350 g. To assess the contrast in normal brain tissue, an intraparenchymal injection of the HMMSN nanorods (0.41 mg mL^−1^) was performed (N = 2). Rats were placed in a stereotaxic frame (Stoelting Co., Wood Dale, IL, USA) under sevoflurane anesthesia. A 1-cm-long midline incision was made in the scalp, beginning midway between the eyes and terminating behind the lambda. A cotton swab was used to clear away the soft tissue covering the skull. A Hamilton syringe (Hamilton; 10 μL) was filled with the nanorods suspension (0.41 mg mL^−1^) dissolved in saline. The syringe was mounted onto the injection pump and the needle was positioned directly over the bregma. The x, y, and z axis coordinates were all set to zero. The needle was then positioned at the entry point, +0.6 mm anterior and −2.9 mm lateral of the bregma to the right. A small cranial burr hole was drilled through the skull at the entry point. The needle was slowly inserted into the basal ganglia to a depth of 5.5 mm below the surface of the skull, and a volume of 10 μL of HMMSN was injected at a rate of 1 μL min^−1^ over 10 min. The needle was left in place for 10 min and then removed at a rate of 1 mm/min to prevent reflux of the injected solution. The same procedure was performed in the left hemisphere, injecting 10 μL of saline. The burr hole was filled with bone wax (Ethicon, Somerville, NJ, USA), and the scalp incision was closed. The rats were placed in an animal box after surgery for recovery in a warm place with access to food.

Total signal intensity was calculated by normalizing the intensity of each injection zone (for saline or HMMSN nanorods solutions) to a physiological tissue intensity area. With the aim to evaluate the potential toxicity effect of the nanorods regarding the administration route, two additional groups were added: Intravenous and intra-arterial route (N = 3/group). For the intravenous administration, 1 mL of HMMNS nanorods (0.41 mg mL^−1^) were gently injected through the right jugular vein of the animal previously exposed with a short incision on the right side of the animal neck. To perform the intra-arterial administration, a small incision was made in the neck of the animal, the muscles were separated to access the common right carotid artery, and the bifurcation area of the external and internal carotid artery. Then, a permanent suture knot was carried out in the right external carotid and pterygopalatine artery, while in the common and internal carotid artery, a transient knot was performed, creating a region with no blood flow to avoid bleeding during manipulation. An incision was made in the external carotid to insert a cannula attached to a syringe containing the HMMNS nanorods suspension. The transient knots of the common and internal carotid were then released, restoring the ascending flow to the brain. The upward flow was maintained for 10 min to stabilize the animal and later, 450 μL was injected for 20 min to avoid volume overload adverse effects due to the increase in blood volume. Once the administration was completed, the cannula was removed, and the external carotid was closed with a permanent knot, the incision in the neck was sutured, and the anesthetized animal was awakened for recovery.

Blood extraction and analysis: A total volume of 1 mL of blood was extracted from the tail vein of the animals and collected in tubes with heparin (BD Vacutainer^®^ Heparin Blood Collection Tubes). The analysis was conducted with a Reflotron^®^ plus (Roche, Basel, Switzerland) by adding 33 µL of the blood sample to reactive strips for Glutamate Oxaloacetate Transaminase enzyme GOT (ref: 10745120202 Roche), Glutamate Pyruvate Transaminase enzyme GPT (ref: 10745138202 Roche), and Creatinine (ref: 10886874202 Roche).

## 3. Results and Discussion

### 3.1. Physicochemical Properties

SBA-15 mesoporous silica nanorods (S15N) were obtained under strongly acidic conditions from the condensation of hydrolyzed silica precursors in the presence of a soft template composed by a mixture of Pluronic P123, a triblock copolymer surfactant, and glycerol, as previously reported [32]. Afterwards, magnetic functionality was added by covalently anchoring [33] magnetite nanoparticles to the S15N surface by coating all with an aminated silica grafting (Figure 1) while the optical response was achieved by linking a dense fluorescent moiety layer (rhodamine B isothiocyanate) to the surface amines of the HMMSN sample.

Structural properties were analyzed by X-ray diffraction at a low angle and wide angle to asses both the small scale order of the mesoporous silica channels and the medium scale order structures, like the magnetite nanoparticles, respectively. Low angle X-ray diffraction patterns of S15N and HMMSN samples, in Figure 2A, show the Bragg diffraction maxima at (100), (110), and (200) planes, corresponding to a highly ordered 2D hexagonal structure (space group P6m), which is characteristic of mesoporous nanostructures [14]. From the position of the (100) diffraction peaks, the unit cell parameters (a_0_) for the S15N and the hybrid HMMSN nanostructures were calculated and are shown in Table 1. In addition, Figure 2B shows the X-ray diffraction pattern at a wide angle range of the hybrid nanostructure HMMSN. The position and relative intensities of crystalline magnetite (Fe_3_O_4_) (inverse spinel structure (JCPDS 19-0629)) allow us to assume that magnetite NPs are attached to the mesoporous silica nanostructures whose amorphous phase is observed as a broad band at the 22–23° 2θ range.

Chemical topology was studied by FITR spectroscopy, which allows an analysis of the vibration bands of the functional groups present at the surface of the samples. In Figure 3, FTIR spectra of mesoporous nanorods, S15N, and their magnetic counterparts, HMMSN, are presented for comparison purposes. In first place, the appearance of a stretching broad band at 1050–1025 cm^−1^ ν(Si–O–Si) and the bending band at 840–780 cm^−1^ indicates the formation of the siloxane matrix. Around 970–940 cm^−1^, ν(Si-OH), the free silanol vibration band of the S15N mesoporous silica, is present, which partially disappears in the case of the HMMSN nanorods. The overlapping with the siloxane broad band is due to a decrease in the amount of available silanol groups present in the functionalized magnetic material. This result is consistent with the formation of a bifunctionalized silanol grafting that may have been anchored to the S15N external surface by condensation of their free silanol groups with dialkoxysilane molecules during the magnetite nanoparticles’ attachment process. In addition, broadened medium bands, between 3400–3100 cm^−1^ and 3550–3200 cm^−1^, are observed, which correspond to the N–H and O–H bonds, respectively. The characteristic C–H vibration bands of the stretching ν(C–H) and bending δ(CH_3_) of the methyl groups are shown at approximately 2900–3000 cm^−1^ and at 1480 cm^−1^, respectively. Finally, the weak peaks at 1630 cm^−1^, 1580 cm^−1^, and 1325 cm^−1^ correspond to the adsorption of water molecules in the material surface δ(H–OH), the scissoring vibration of the N–H bond, and the stretching vibration of the C–N bond, respectively.

Moreover, HMMSN nanorods were analyzed by elemental chemical analysis (ECA) to determine the nitrogen content present in the sample. The experimental weight percentage of functionalization was calculated assuming that all nitrogen found in the sample proceeded from the hydrolysed 3-aminopropyl-methyl-diethoxysilane molecules, which was used to trap the magnetite NPs onto the external S15N surface by the formation of a graft (Figure 1). Considering this, the so obtained weight percentage, 22.5%, confirms that the surface was successfully modified by a large number of organic molecules. In addition, this value highlights the efficiency of the synthetic single step double procedure: Anchoring magnetite NPs and functionalizing the surface at the same time.

The fluorescent properties of the rhodamine coated nanorods were studied by means of fluorescence spectrometry and confocal microscopy. Figure 4A presents the photoluminescence spectra of the rhodamine functionalized HMMSN nanorods, showing that the excitation has a maximum located at λ^max^
_ex_ = 560 nm, for which a maximum emission peak is observed at λ^exc^
_em_ = 580 nm in concordance with the fluorescence excitation/emission pattern of rhodamine (λ_ex_ = 543 nm and λ_em_ = 580 nm). Fluorescent confocal micrographs under a single 552 nm laser excitation are shown in Figure 4B, where the intense red color emission spots, corresponding to the fluorescent HMMSN nanorods, are depicted as being evenly distributed. This optical functionality added to the silica surface without modifying the structural, textural, or physical properties endows the nanorods with additional fluorescent tagging for biological imaging.

The textural properties of S15N and HMMSN were assessed by performing N_2_ adsorption isotherms as can be observed in Figure 5A, which shows a characteristic type-IV isotherm of SBA-15 mesoporous materials with an ordered cylindrical pore arrangement with open ends. Both curves show a characteristic H1-type hysteresis cycle [34] (parallel adsorption and desorption branches) with different areas. The smaller HMMSN area corresponds to the partial clogging of the pores by the magnetite NPs, which reduce the availability of channels.

The textural properties (pore size distribution and specific surface area) were estimated from the N_2_ adsorption isotherms and are plotted in Figure 5B. It is evident that the HMMSN hybrid nanorods present a similar pore size (around 9 nm) and a narrower pore size distribution than the precursor mesoporous matrix. This result highlights the efficiency of the present functionalization procedure, which, besides alternative routes that block the pores [35,36], it is also able to provide a large degree of surface modification while preserving the textural properties of native S15N. In Table 1, a comparison of the textural properties between native S15N and hybrid HMMSN is shown for both materials, including a large specific surface area, which endows them with a high capacity for loading moieties of therapeutic interest.

The morphology of native S15N and hybrid HMMSN nanorods was studied with the help of transmission and scanning electron microscopy. In Figure 6, SEM and TEM images, on the top and bottom, respectively, can be observed for S15N mesoporous silica on the left, and hybrid HMMSN nanorods on the center and right in different locations and with different magnifications. TEM images of the S15N matrix reveal a hexagonal structure with a cylindrical ordered pore disposition with a diameter estimated from the images to be 8.5 nm, in good agreement with the value estimated by the BJH method of 8.177 nm. On the other hand, the SEM images show that the S15N matrix is mainly composed of small particles with a uniform rod morphology. In addition, the presence of magnetite nanoparticles anchored on the S15N ceramic surface can be clearly identified for the HMMSN nanorods, showing a well-defined mesoporous structure with NPs homogeneously distributed around it. The average pore diameter estimated from these micrographs was 9.1 nm, higher than the value estimated from the BJH method of 7.40 nm, which means that the anchoring procedure may have slightly distorted the ordered structure of the mesoporous silica, as has already been observed before [33]. This increase in pore diameter partly explains the observed increase in the pore volume.

The size distribution statistics of these rodlike particles (Figure 7) display an average width of 92 ± 25 nm, and a length in the range between 300 and 540 nm, similar to those found by Johansson [37]. Rod-like morphology can be tailored by controlling the acid mixture (HCl/H_3_PO_4_), which is a key factor in the production of nanorods [14].

The ^29^Si solid-state MAS NMR spectrum shows the identified signals corresponding to different Q^n^ and D^n^ silicon species [38] present in the HMMSN material associated with the relative amounts of silica frameworks and their corresponding functionalization.

Figure 8 shows the ^29^Si solid-state MAS NMR spectrum of the HMMSN sample. The most representative chemical shifts for the local silicon environments in the HMMSN nanorods are D^1^ (δ = −20.51 ppm), D^2^ (δ = −26.99 ppm), D^2^ (δ = −31.40 ppm), and Q^4^ (δ = −114.24 ppm). The main difference observed in Table 2 is related to the number of oxygen atoms connected to each silicon atom, which forms the building units. Thus, the replacement of O atoms by aminopropyl and methyl groups caused substantial high-frequency (downfield) shifts, therefore, the chemical shifts of the siloxane centers are sensitive to neighbor effects in the chain structure, thus revealing the microstructure of the material.

The most intense peaks are located between −18 and −32 ppm are related to D^n^ siloxane centers, coinciding with that reported by Wiench et al. [39], and thus corroborating the functionalization and anchoring of magnetite nanoparticles onto the silica surface. The results are summarized in Table 2. Additionally, the areas under the curve were quantified to approximately define the bands associated with each ^29^Si chemical shift of the HMMSN material.

Figure 9 shows the room temperature hysteresis loop of the functionalized nanorods as a function of the applied magnetic field (up to 10 kOe). The material exhibits an excellent magnetic response normalized to the magnetite mass with clear superparamagnetic (SPM) behavior. Compared to the saturation magnetization value obtained with other similar magnetic mesoporous nanostructures previously reported by our group [33] and by other research groups [40,41,42,43], this is higher and similar to some functionalized magnetite nanoparticles [44,45]. The magnetization M vs. magnetic field H cycle is slightly shifted to negative magnetic fields; that may be related to magnetostatic interactions [46] or exchange anisotropy [47] since magnetite nanoparticles are attached to the external mesoporous nanorod and thus cannot be treated as if each nanoparticle were a pure dipole.

### 3.2. Relaxivity and Brain MRI Contrast of the HMMSN Nanorods

The HMMSN nanorods relaxivity was assessed by magnetic resonance, showing values of 258.94 mM^−1^ s^−1^ and 324.68 mM^−1^ s^−1^ for T2 and T2* sequences, respectively (Figure 10). In this sense, the iron content (Table 1) provides enough contrast for adequate imaging compared to other commercial contrast agents [48].

The HMMSN nanorods suspension was injected in the right hemisphere of male Sprague-Dawley rats; and a vehicle injection of saline was performed in the left hemisphere (Figure 11A). Figure 11B shows that the T2 signal intensity in the nanoparticles injection area decreased to around 58%, generating an hipointensity area with increased contrast compared to a normal tissue intensity region, while the vehicle signal intensity only decreased by 5.7%. The T2* signal intensity decreased by 52% while the vehicle group only around 15%. Nanoparticles concentrated in the injection spot enhance the contrast with normal brain tissue in both the T2 and T2* sequences.

### 3.3. In Vitro Cell Mortality by LDH Toxicity Assay and Prussian Blue Stain of the HMMSN Nanorods

Nanotechnology allows the achievement of high levels of sophistication and personalization for different applications, like diagnosis, therapy, or even both. One of the biggest challenges has always been to deal with the brain; in brief, overcoming the blood brain barrier and reaching an adequate concentration of nanoparticles to produce the desired effect without causing any side damage have always been the main issues. In this regard, endothelial cells are the primary component that perform most interactions with the nanoparticles suspension after systemic administration, impairing the physiological structure of the cells, damaging the membrane, and altering the integrity and permeability of the layer [49,50]. According to this, it will be mandatory to assess the potential toxicity derived from different interactions of the new materials and covers involved in this process. This work is focused on possible future applications in the brain area; in this sense, brain microvasculature cells (Bend.3) were selected due to their strong relationship with brain permeability and their pivotal role in the neurovascular unit.

Endothelial cells were cultured with the HMMSN nanorods for 24 h, showing a significant mortality only when the highest amount of 200 µg was used. Regarding the samples with 100, 50, and 25 µg amounts, they showed no significant mortality compared to the control group (Figure 12).

The uptake was evaluated in a qualitative way through a Prussian Blue stain of the nanorods, as shown in the Figure 13; enough nanorods were taken up by cells to cover most of the citoplasmatic portion. This elevated uptake is probably due to the presence of amino functionalized nanorods and the surface affinity of the cells to these groups, which probably induce a strong attraction, thus increasing the binding/uptake. However, the presence of methyl and silanol groups contribute by having a negative charge in the pH conditions of deionized water as was corroborated by the measured Z potential, −18.63 mV, which is highly convenient since cationic nanoparticles are usually associated with high toxicity as Fröhlich demonstrated [51]. In summary, only the highest concentration shows a significant toxicity due to the massive presence of NRs in the cells, which is evidence of the optimal biocompatible properties of mesoporous nanorods with brain endothelial cells.

### 3.4. Brain MRI Evaluation after Intravenous and Intra-Arterial Injections and In Vivo Toxicity of the HMMSN Nanorods

Toxicity due to particle-cell interaction is not the only source of potential damage in nanoparticles treatments; in in vivo models, mechanical damage is also important as anastomosis in diverse organs reduces vessel caliber trapping of the particles, causing occlusions that could lead to ischemic lesion or could damage the integrity of the vessels’ structure, leading to a possible hemorrhagic transformation. Another aspect considered in this work is the administration route. After being injected into the external carotid artery, the first tissue with which particles have contact with is the cerebral microvasculature. Due to its advantageous ascending flow, this is the optimal route to observe any effects on the cerebral area without passing through the rest of the organs with a consequent loss of nanoparticles. On the other hand, venous administration through the jugular vein, where a descending flow transports the nanoparticles to the heart and then to the rest of the body, disseminating them to different organs of the animal, allows us to evaluate their effect [52].

Based on this, a total of 1.4 mg kg^−1^ of mesoporous nanoparticles were administrated in two different groups: Venous administration group and arterial administration group. Animals were followed by MRI for 14 days to evaluate any possible associated damage in the brain. No ischemic or hemorrhagic damage were observed through the magnetic resonance images in any group (Figure 14), and there was also no evidence of nanoparticles in the brain, which indicates no apparent BBB disruption due to nanoparticle interaction. The particle accumulation effect is especially relevant in well-vascularized organs, where any malfunction leads to several impairments in the animal [53]. Based on this, we have focused on the particle effect over the liver and kidney, two commonly damaged organs due to particle accumulation or tissue disruption. For this purpose, GOT, GPT, and creatinine were analyzed in the blood at different time points for 14 days [54]. An increase of GOT and GPT enzymes in serum is related to liver damage [55,56], while creatinine levels are associated with kidney malfunction [57]. No increments in GOT and GPT activity were observed in any of the groups compared to baseline levels. Creatinine levels were always under the detection limit (<0.5 mg dL^−1^) and no increments were observed in any of the groups or time points. On the other hand, all rats were observed for 1 month to control their activities in case of any respiratory failure or lethargic behavior, and all showed normal activity.

Different parameters, such as charge, size, or shape, influence the cell–particle interaction [58], and can increase tropism determined by the cellular type or facilitate its clearance and blood circulation times. For instance, a positive charged particle increases the electrostatic attraction to the superficially negative cells, which causes an increase in the toxicity response, while a negative charge causes a greater phagocytosis of macrophages. Similar dualities are associated to size [59] and shape [60], with rod-like particles similar to that described in this work showing less uptake due to the extra energy needed by the cell to wrap the particle compared to spherical shapes [61]. However, when coated with antibodies or similar strategies for active targeting, rods show higher binding than spherical shapes [62]. This makes nanoparticles a highly versatile tool that allows modifications to be performed on the molecular structure to enhance some interactions and decrease others according to the target requirements.

This work has demonstrated a first approach for the safe use of rod shaped magnetic mesoporous nanorods at the concentrations described above, with no potential damage to endothelial brain cells, even showing an apparently high level of uptake. Magnetic resonance imaging showed no brain damage for 14 days, and no liver or kidney impairment, which evidences its safe use both in vitro and in vivo, overcoming toxicity impairments, and allowing its future use as a harmless contrast agent or therapeutic tool.

## 4. Conclusions

Multimodal mesoporous nanorods, with controlled morphology and size, were developed and studied, showing good magnetic performance for MRI contrast activity, intense fluorescent response, large specific surface area for high capacity loading, and a versatile surface for grafting biologically active moieties

In addition, an extended biocompatibility study was performed by analyzing the cell toxicity and organ status of male Sprague-Dawley rats after being injected with samples of nanorods by different administration routes (intravenous and intra-arterial), confirming that animals remained unaltered after the treatment with no organ disability.

In addition, intraparenquimal injection of nanorods corroborated their capacity as contrast agents in the brain due to a higher T2 signal intensity than usual, thus providing enhanced contrast. Likewise, using LDH assay, the Prussian Blue stain displayed a large nanorods’ uptake that covered most of the cell cytoplasm, showing that only the higher concentration presented toxicity with respect to the control group.

This preliminary physicochemical and biological in vitro/in vivo study shows that magnetic mesoporous silica nanorods present a combination of properties that make them suitable candidates to be tested as multifunctional nanoreservoirs in brain endothelium applications.

## Figures and Tables

**Figure 1 nanomaterials-09-00449-f001:**
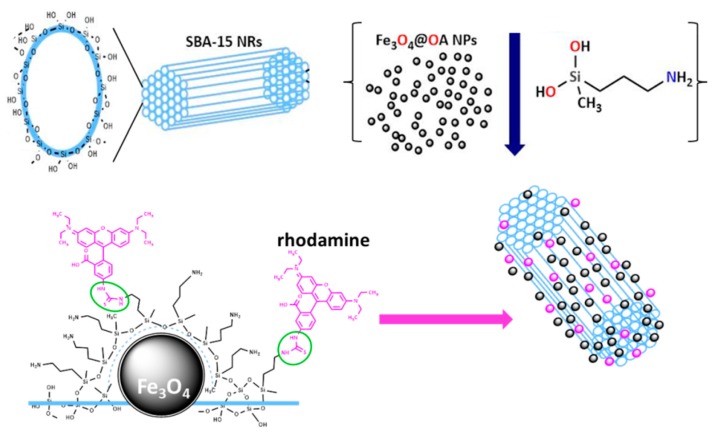
Scheme of the ordered mesoporous silica ceramic with Si–OH groups exposed on the surface (top). Anchoring of magnetite nanoparticles by adding functional ethoxysilane molecules allows the formation of the HMMSN hybrid mesoporous magnetic nanorods. Fluorescent dye (rhodamine) is further attached on top.

**Figure 2 nanomaterials-09-00449-f002:**
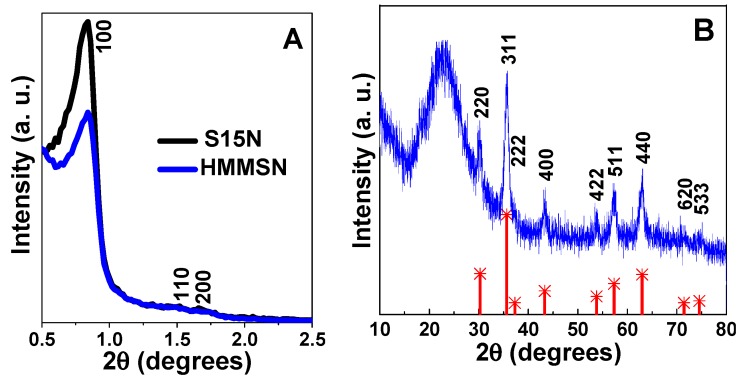
(**A**) Low-angle XRD patterns of the S15N mesoporous silica and HMMSN hybrid magnetic mesoporous nanorods and (**B**) the XRD pattern of HMMSN hybrid magnetic mesoporous nanorods (blue), with the theoretical XRD pattern of magnetite (red) for comparison purposes.

**Figure 3 nanomaterials-09-00449-f003:**
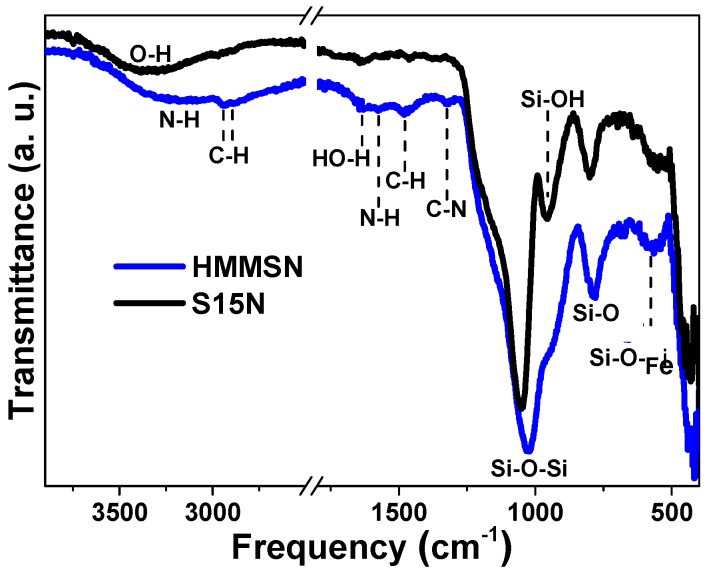
FTIR spectra of the S15N mesoporous silica matrix and HMMSN hybrid magnetic mesoporous nanorods.

**Figure 4 nanomaterials-09-00449-f004:**
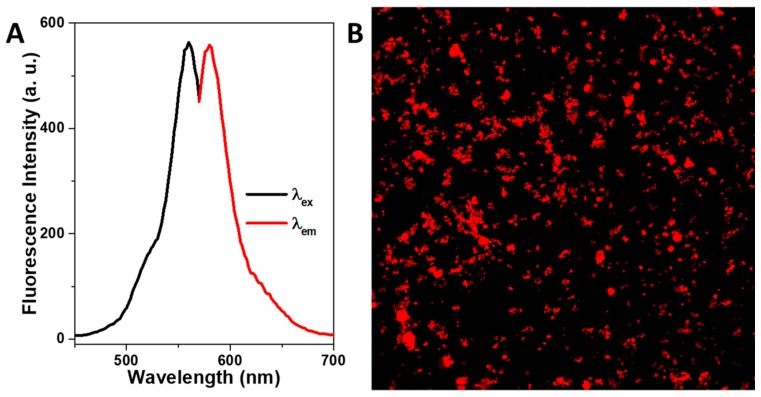
(**A**) Photoluminescence spectra and (**B**) confocal fluorescence images of HMMSN hybrid magnetic mesoporous nanorods.

**Figure 5 nanomaterials-09-00449-f005:**
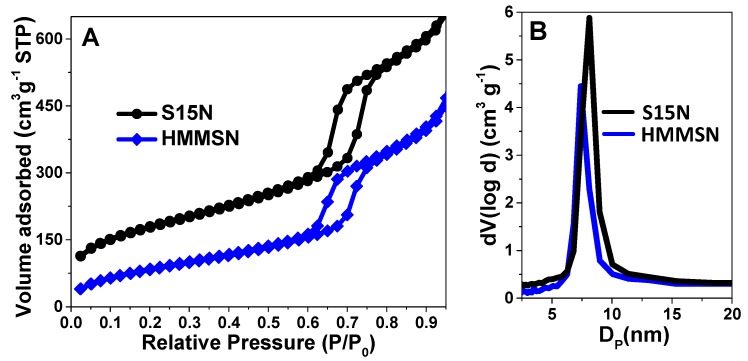
(**A**) N_2_ sorption isotherms and (**B**) pore size distribution of the SBA-15 matrix and HMMSN hybrid magnetic mesoporous nanorods obtained from the BJH method.

**Figure 6 nanomaterials-09-00449-f006:**
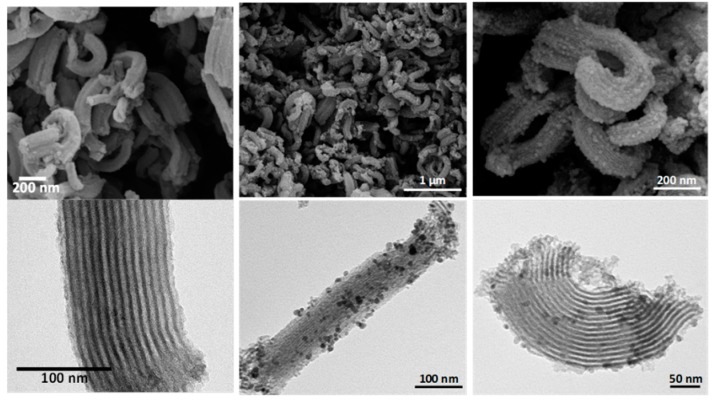
SEM (**top**) and TEM (**bottom**) micrographs of (**left**) S15N mesoporous silica and (**center** and **right**) HMMSN hybrid magnetic mesoporous nanorods in different locations and with different magnifications.

**Figure 7 nanomaterials-09-00449-f007:**
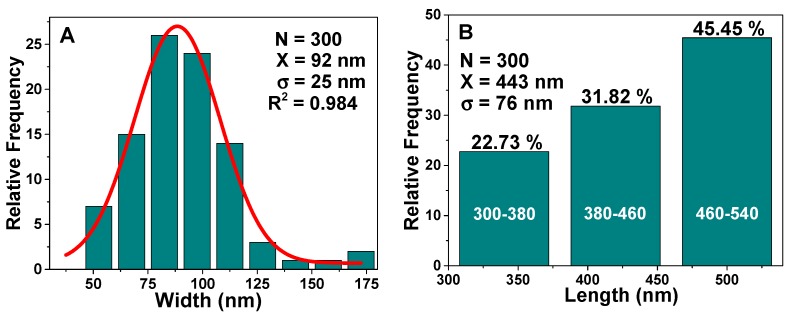
(**A**) Width and (**B**) length distributions of the S15N mesoporous silica nanorods. (**A**) N is the sampling, X and σ are the center and width of the Gaussian distribution, while for (**B**), X and σ are obtained from the arithmetic mean.

**Figure 8 nanomaterials-09-00449-f008:**
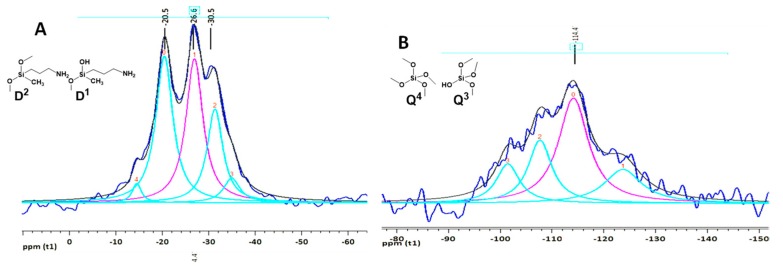
^29^Si solid-state CP-MAS NRM spectra of HMMSN hybrid magnetic mesoporous nanorods with fittings of (**A**) D^n^ siloxane centers and (**B**) Q^n^ siloxane centers.

**Figure 9 nanomaterials-09-00449-f009:**
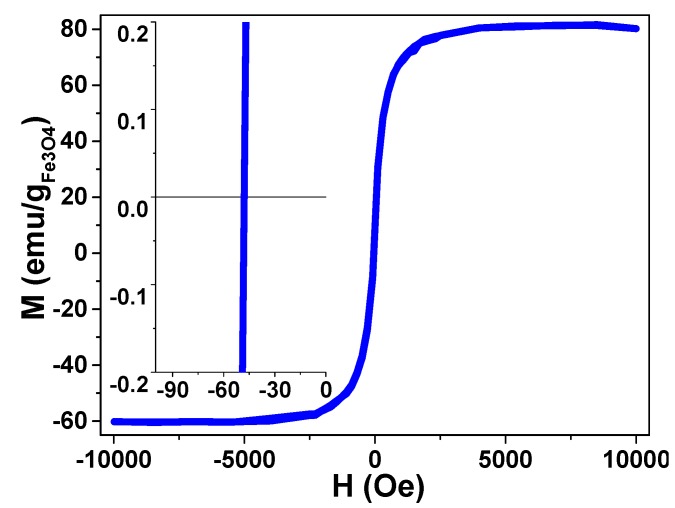
Magnetic hysteresis loop from 10 to −10 kOe of HMMSN hybrid magnetic mesoporous nanorods up to 10 kOe, measured at room temperature, and a magnification of the low field region, revealing an almost absent coercive file.

**Figure 10 nanomaterials-09-00449-f010:**
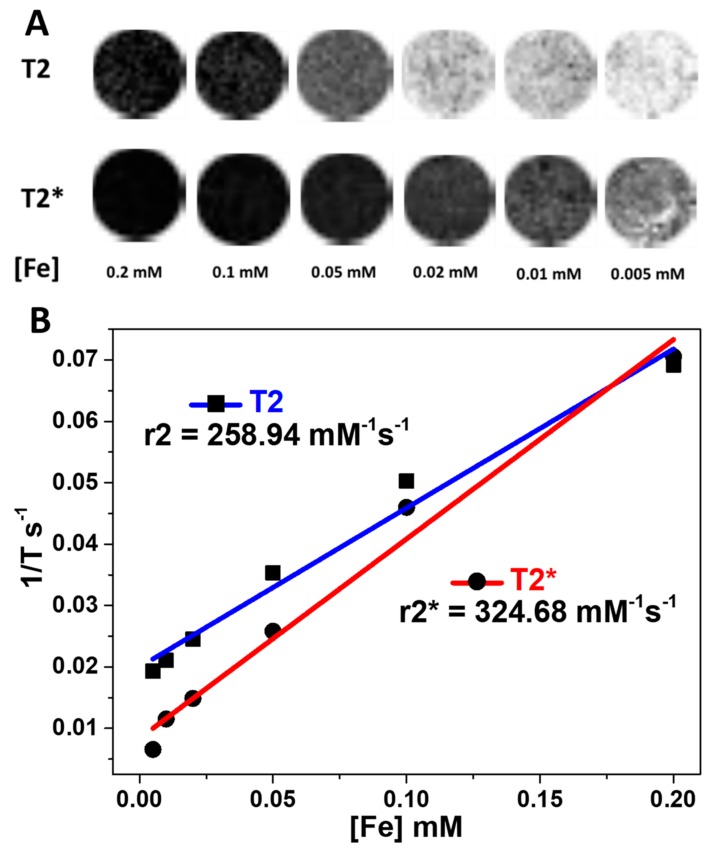
(**A**) T2 and T2* weight images of an agar phantom template, with crescent concentrations of Fe coupled to HMMSN hybrid magnetic mesoporous nanorods. (**B**) Relaxivity values obtained with T2 and T2* weight images.

**Figure 11 nanomaterials-09-00449-f011:**
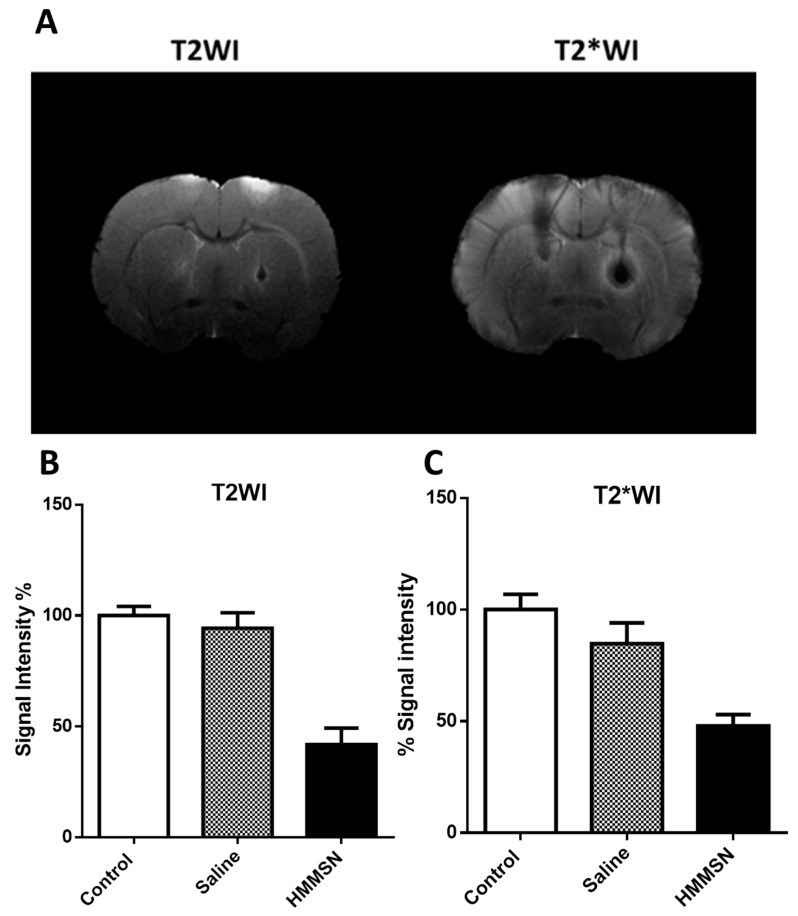
(**A**) T2 and T2* weight images after intraparenchymal injection of 10 µL 0.9% NaCl saline (left hemisphere) and 10 µL of a suspension (0.41 mg/mL) of HMMSN hybrid magnetic mesoporous nanorods in saline (right hemisphere). (**B**,**C**) % signal intensity is decreased in the injection area compared to the control for normal tissue intensity in T2 and T2* weight images, respectively. (N = 2/group).

**Figure 12 nanomaterials-09-00449-f012:**
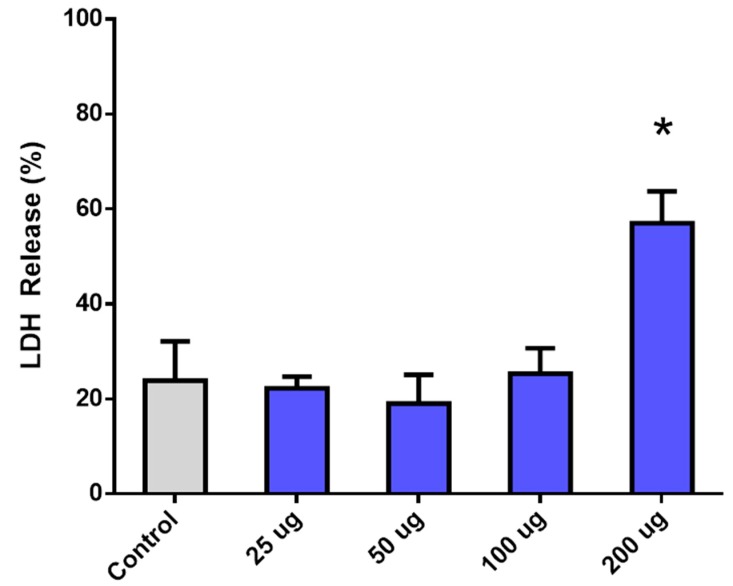
Cell viability assay representing the percent of LDH release of Bend.3 endothelial cells treated with increasing amounts of HMMSN hybrid magnetic mesoporous nanorods. Values are normalized to a positive control for cell death. (N = 8/groups, * = *p* < 0.005 compared to the control group).

**Figure 13 nanomaterials-09-00449-f013:**
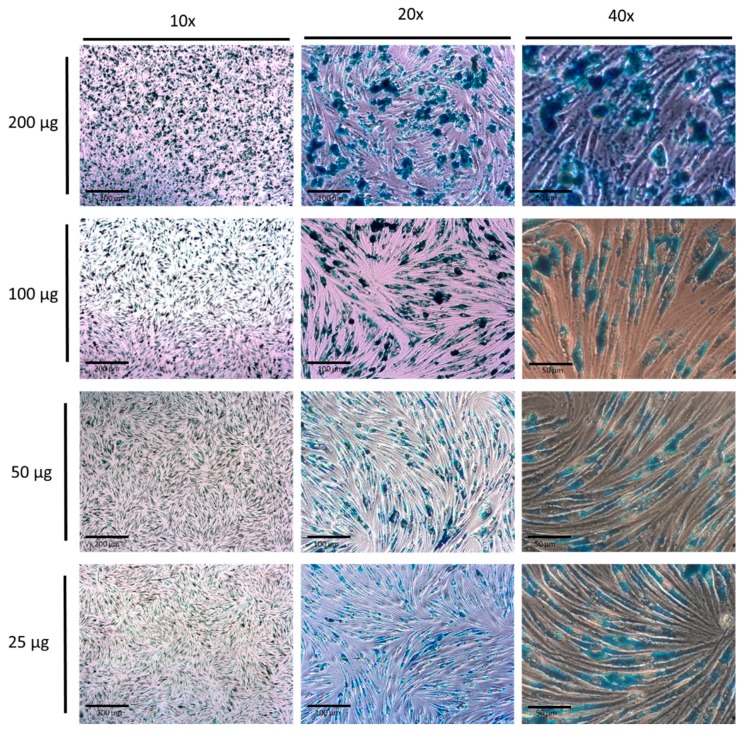
Prussian Blue stain of HMMSN hybrid magnetic mesoporous nanorods after 24 h of incubation with Bend.3 endothelial cells. Scale bars: 10× = 200 µm, 20× = 100 µm, and 40× = 50 µm.

**Figure 14 nanomaterials-09-00449-f014:**
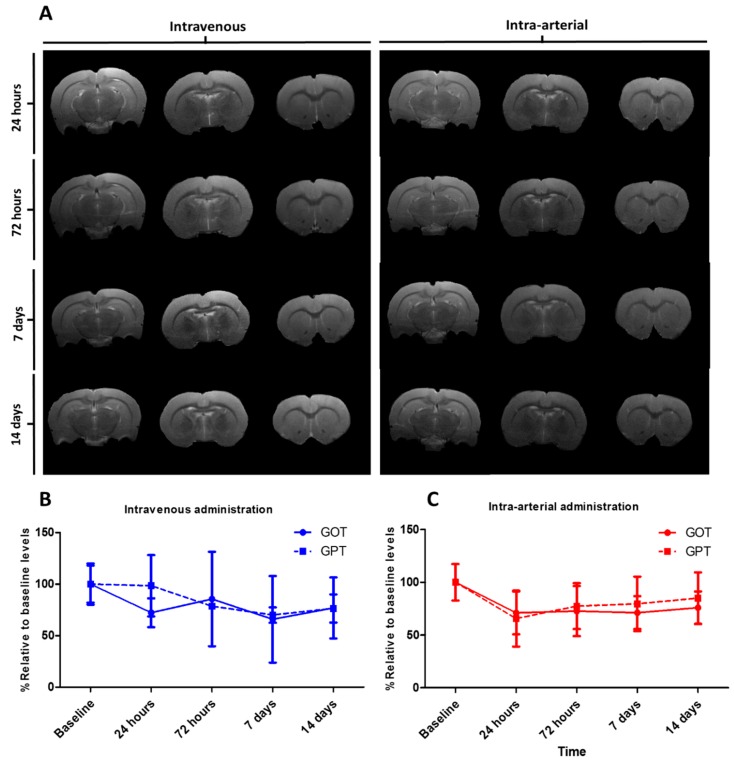
(**A**) T2 weight images after intravenous and intra-arterial administration of 1.4 mg/kg of HMMSN hybrid magnetic mesoporous nanorods. (**B**,**C**) GOT and GPT levels normalized to baseline at different time points after intravenous and intra-arterial administrations, respectively. (N = 3/group).

**Table 1 nanomaterials-09-00449-t001:** Textural and structural values of the SBA-15 matrix and HMMSN hybrid magnetic nanocomposite. Surface area (S_BET_), total pore volume (V_p_), pore diameter (D_BJH_), unit cell parameter (a_0_), thickness of the mesopores walls (t_wall_ = a_0_ − D_BJH_), and iron oxide weight percentage (W_mag_), expressed as Fe_3_O_4_, determined by flame atomic absorption spectroscopy, FAAS.

Samples	S_BET_ (m^2^ g^−1^)	V_p_ (cm^3^ g^−1^)	D_BJH_ (nm)	d_10_ (nm)	a_0_ (nm)	t_wall_ (nm)	W_mag_ (%)
**S15N**	617.10	1.78	8.18	10.52	12.15	4.03	0.00
**HMMSN**	318.97	1.82	7.40	10.51	12.14	4.77	7.60

**Table 2 nanomaterials-09-00449-t002:** Chemical shift and silicon environments of the HMMSN hybrid magnetic mesoporous nanorods.

Chemical Shift (ppm)	Silicon Environment	Area Under a Curve (%)
−123.80	Q^4^	5.26
−114.24	Q^4^	12.89
−107.77	Q^4^ + Q^3^	6.12
−101.55	Q^3^	3.78
−35.01	D^2^	4.18
−31.40	D^2^	15.12
−26.99	D^2^	25.22
−20.51	D^1^	25.59
−14.58	D^1^	1.84

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
