# Peer review of "Multifunctional Superparamagnetic Stiff Nanoreservoirs for Blood Brain Barrier Applications"

_nanomaterials, 2019, doi:10.3390/nano9030449_

Reviewer 1 Report

Manuscript prepared casually.

Unfortunately, the line numbering is lost after page 5. This complicates the review. Line failure after line 166 is also observed. The numbering of the figures is also broken, the reference to Figure 4 comes immediately after Figure 1. It is necessary to reformat the numbering of both lines and figures.

There are many figures in the manuscript, Figure 5 can be removed without loss of important information. Alternatively, the Figures 1 and Figure 4 can be combined, and Figures 7 and Figure 9 too. Figure 10 can be placed inside figure 11.

Figure 2. (A) Low-angle XRD patterns of the S15N mesoporous silica and HMMSN hybrid magnetic mesoporous nanorods and (B) XRD pattern of HMMSN hybrid magnetic mesoporous nanorods, with the theoretical XRD pattern of magnetite (drop lines) for comparison purposes.

Have to be

Figure 2. (A) Low-angle XRD patterns of the S15N mesoporous silica and HMMSN hybrid magnetic mesoporous nanorods and (B) XRD pattern of HMMSN hybrid magnetic mesoporous nanorods (blue), with the theoretical XRD pattern of magnetite (red) for comparison purposes.

Figure 5. A) PL spectra and B) Confocal fluorescence images of HMMSN hybrid magnetic mesoporous nanorods.

What is PL Spectra? What resolution or scale is used in microscopy?

Figure 8. (A) Width and (B) Length distributions of the S15N mesoporous silica nanorods. N is the sampling, X and σ are the center and width of the Gaussian distribution, while for 6B, X and σ are obtained from the arithmetic mean.

What is 6B?

207 conditions but only in presence of a HCl solution (Figure S1), in this sense, the acid mixture

Where is Figure S1?

Line 209-210

The description of figure 9 is omitted in the text of the manuscript, it should be added.

221 14A). Figure 14B shows that T2 signal intensity in the nanoparticles injection area is decreased

Have to be

14A). Figure 14B and 14C show that T2 signal intensity in the nanoparticles injection area is decreased

223 The description of Figure 15 should be after the second paragraph where there is a link to it. I did not find the Figure S2 in manuscript.

226 The description of Figure 17 should be after line 228

Manuscript must be carefully reworked.

Author Response

Uploaded file with responses to Referee 1

Reviewer 2 Report

see attached file

Author Response

File uploaded

Reviewer 3 Report

Please note that my review is limited to the synthetic procedure and the structure of the obtained magnetic nanomaterials.

The different characterizations clearly show that SBA-15 silicas contains the MNPs on their external surface. It is however more difficult to conclude on the exact nature of the interaction between the MNPs and the silica nanorods. Indeed, the synthetic procedure which is given is based on a paper (ref 33) where TEOS was added to an Igepal microemulsion containing the SBA-15 nanorods and the oleic acid coated-MNPs. This previous work was based on the work of Ana B. Davila-Ibanez et al. (ref 69 in ref 33), which had confirmed that, when TEOS is added to oleic acid coated-MNPs alone in Igepal microemulsion, core-shell silica coated MNPs are obtained. In the conditions of ref 33, we can reasonably consider that the interaction between the silica coated MNPs and the silica surface of SBA-15 is due to the formation of siloxane bridges between the silica shell of the MNPs and the silica surface of the SBA-15.

However, here, TEOS is replaced by aminopropylmethyldimethoxysilane (APMDMS). This compound is an unusual coupling agent, with only to polymerizable methoxy groups. First, I don’t understand why this compound is used instead of aminopropyltrimethoxysilane (APTMS) which have three polymerizable groups, or instead of a mixture of APTMS and TEOS. Second, the forming of a hybrid shell around the MNPs is not guaranteed with APMDMS (this compound being less hydrophilic and less polymerizable than APTMS and/or TEOS, it may have a totally different behaviour). Therefore, although NMR and fluorescent measurements prove that APMDMS is grafted on SBA-15, simultaneously to the MNPs, it is unclear if this coupling agent is responsible for the anchoring of the MNPs on the SBA-15, and if a regular shell of APMDMS is formed around the MNPs.

It would be interesting that the authors explain why they chose APMDMS, and give some evidences that this coupling agent is responsible for the anchoring of the MNPs on the SBA-15, through the forming of a shell around the MNPs (for example, they may study the formation of core-shell APMDMS coated MNPs, by performing the same procedure in the absence of SBA-15).

Author Response

The authors acknowledge the reviewers work, for their suggestions and comments,  that we think have clarified  the manuscript. Please find enclosed below the answers to this second review .

Referee 3 . Please note that my review is limited to the synthetic procedure and the structure of the obtained magnetic nanomaterials.   The different characterizations clearly show that SBA-15 silicas contains the MNPs on their external surface. It is however more difficult to conclude on the exact nature of the interaction between the MNPs and the silica nanorods. Indeed, the synthetic procedure which is given is based on a paper (ref 33) where TEOS was added to an Igepal microemulsion containing the SBA-15 nanorods and the oleic acid coated-MNPs. This previous work was based on the work of Ana B. Davila-Ibanez et al. (ref 69 in ref 33), which had confirmed that, when TEOS is added to oleic acid coated-MNPs alone in Igepal microemulsion, core-shell silica coated MNPs are obtained. In the conditions of ref 33, we can reasonably consider that the interaction between the silica coated MNPs and the silica surface of SBA-15 is due to the formation of siloxane bridges between the silica shell of the MNPs and the silica surface of the SBA-15.

A) However, here, TEOS is replaced by aminopropylmethyldimethoxysilane (APMDMS). This compound is an unusual coupling agent, with only to polymerizable methoxy groups. First, I don’t understand why this compound is used instead of aminopropyltrimethoxysilane (APTMS) which have three polymerizable groups, or instead of a mixture of APTMS and TEOS.

B) Second, the forming of a hybrid shell around the MNPs is not guaranteed with APMDMS (this compound being less hydrophilic and less polymerizable than APTMS and/or TEOS, it may have a totally different behaviour). Therefore, although NMR and fluorescent measurements prove that APMDMS is grafted on SBA-15, simultaneously to the MNPs, it is unclear if this coupling agent is responsible for the anchoring of the MNPs on the SBA-15, and if a regular shell of APMDMS is formed around the MNPs.

 C) It would be interesting that the authors explain why they chose APMDMS, and give some evidences that this coupling agent is responsible for the anchoring of the MNPs on the SBA-15, through the forming of a shell around the MNPs (for example, they may study the formation of core-shell APMDMS coated MNPs, by performing the same procedure in the absence of SBA-15).

A.                  After Journal Materials Chemistry B  2017,5, 9395, was published, we decided to modify the method in order to optimize the properties of the so obtained magnetic mesoporous nanostructures. For this reason, we tested three different  alkoxilane coupling agents [APMDMS(3-Aminopropyl(diethoxy)methylsilane), with 2 polymerizable silanol  groups; APTES, (3-Aminopropyl)triethoxysilane), with 3 polymerizable silanol groups; DAMO, (N-[3-(Trimethoxysilyl)propyl]ethylenediamine, 3 polymerizable silanol groups],  to anchor MNPs on the mesoporous silica nanostructures surface and, try at the same time to  functionalize it with  amine groups.

All the so obtained nanocomposites were prepared under the same  conditions (except the coupling agent), washed several times, and  finally characterized by TGA, IR spectroscopy, BET, etc. The results of this study are compiled in the table, showing that APMDMS-based nanocomposite had beter global textural properties and a magnetite content similar to the APTES based material, and both similar to the one reported with TEOS, around 7%.  The nanocomposite prepared with DAMO, although being a 3 polymerizable silanol group, was clearly below the grafting ability of APMDMS (2-polymerizable silanol group).

Samples

SBET

(m2 g-1)

Vp

(cm3 g-1)

Wmag (%)

APMDMS*

318.97

1.82

7.60

APTES

289.44

1.515

7.60

DAMO

285.14

1.39

5.25

TEOS

315.13

1.098

7.71

Table: compilation of properties (* APMDMS  was choose to perform HMMSN) 

                        Figure:  left) small angle XRD and right) BET pore diameter determination.

Therefore, the reason we choose the procedure based on  APMDMS although it has only two 2 polymerizable silanol groups was due  to the good textural  properties of the resulting nanocomposites

B. We agree with the Referee  that APMDMS is less reactive than TEOS or  APTES, which have 3 silanol reacting groups, and that similarly to bifunctional monomers, which are only able to create linear polymers or loops,  it is not an effective compound to create a dimensional network. However, in the present case, the  three different species involved (magnetite NPs with -OH, mesoporous silica with -SiOH, and the APMDMS disilanol groups), can condensate into siloxane bridges,  creating, not a layer, but the grafting of magnetite NPs on the mesoporous silica surface (maybe also with the creation of some intermediate  siloxane bridge loose loops).   The fact that magnetite NPs remain attached to the mesoporous silica surface, during many  manipulations and procedures we have applied, like centrifugation, bioconjugation, etc, seems to us reasonable to think that APMDMS is responsible of this anchoring.

However, we agree with the Referee that the term layer, which one relates to the idea of a homogenous and appreciable network phase surrounding another material phase, was incorrectly   used by us.

In fact, we have corrected in the text, the inexact term layer by grafting, which is a much more realistic description of the functionalization of magnetite NPS with the grafted APMDMS, whose ending amine groups are involved in the effective  coupling of  fluorescent Rhodamine molecules. 

C. For our work in  J.Chem. B, we performed an initial study, for determining the effect of adding different concentrations of TEOS to obtain Fe3O4@SiO2.  For the lowest  dilution rate 1:5, we cannot observe a physical layer in the TEM images, but only a  grafting, of the alkoxilane species. We choose this dilution  rate  (1:5) for all the tested coupling agents studies, and specifically also in the present manuscript,   in order to preserve mesoporous silica  textural properties as much as possible after the anchoring procedure.  However, we did not perform a microemulsion study using APMDMS as the silica precursor, to develop single core Fe3O4@SiO2, since our aim was to obtain a stable anchoring of magnetite NPs on the mesoporous silica surface.

   Since the deadline to provide answers for this second review   is very tight ( 3days) we cannot perform the suggested  experimental study to be included in the present  correction.

Round  2

Reviewer 1 Report

This time, the authors have done a lot for the manuscript to be accepted for publication. They answered all the questions and corrected the text and the figures. In this form, the manuscript can be accepted for publication. Methodological aspects and the results will be of interest to readers.

Author Response

The authors want to acknowledge the possitive evaluation of Referee 1 after our first round of corrections.

Reviewer 2 Report

Dear Authors , as opposed to your J.Mater.Chem.B,2017,5,9395 in the line 133 to 142 there is no TEOS added to allow the formation of the silica layer between fe3o4 and your S15N.

A disilanol cannot do the job.

Author Response

Please find enclosed the answers to the questions posed in this second review and In addition,  we have quoted in the text , reference 14, our  recently accepted manuscript related to the sustained delivery effect of mesoporous silica, as suggested  previously.

Ref 14     Z. Vargas-Osorio, Asteria Luzardo-Álvarez Y. Piñeiro, C. Vázquez-Vázquez, J. Luis Gómez-Amoza, José Blanco-Méndez, Francisco J. Otero Espinar and José Rivas, , ACS Omega http://dx.doi.org/10.1021/acsomega.8b03676 accepted for publcation, Feb 2019.

Referee 2 - A)  Dear Authors, as opposed to your J.Mater.Chem.B,2017,5,9395 in the line 133 to 142 there is no TEOS added to allow the formation of the silica layer between fe3o4 and your S15N.  B) A disilanol cannot do the job.

A.                       After Journal Materials Chemistry B  2017,5, 9395, was published, we decided to modify the method in order to optimize the properties of the so obtained magnetic mesoporous nanostructures. For this reason, we tested three different  alkoxilane coupling agents [APMDMS(3-Aminopropyl(diethoxy)methylsilane), with 2 polymerizable silanol  groups; APTES, (3-Aminopropyl)triethoxysilane), with 3 polymerizable silanol groups; DAMO, (N-[3-(Trimethoxysilyl)propyl]ethylenediamine, 3 polymerizable silanol groups],  to anchor MNPs on the mesoporous silica nanostructures surface and, try at the same time to  functionalize it with  amine groups.

All the so obtained nanocomposites were prepared under the same conditions (excepting the coupling agent), washed several times, and  finally characterized by TGA, IR spectroscopy, BET, etc. The results of this study are compiled in the table, showing that APMDMS-based nanocomposite had better global textural properties, than the rest,  and a magnetite content similar to the APTES and TEOS (reported in J. Mat. Chem.B) around 7%.  The nanocomposite prepared with DAMO, although being a 3 polymerizable silanol group, was clearly below the grafting ability of APMDMS (2-polymerizable silanol group), showing a lower content of anchored magnetite.

Samples

SBET

(m2 g-1)

Vp

(cm3 g-1)

Wmag (%)

APMDMS*

318.97

1.82

7.60

APTES

289.44

1.515

7.60

DAMO

285.14

1.39

5.25

TEOS

315.13

1.098

7.71

Table: compilation of properties (* APMDMS  was choose to perform HMMSN) 

                        Figure:  left) small angle XRD and right) BET pore diameter determination.

Therefore, since textural properties of APMDMS based nanocomposites were globally better than that of the other obtained nanocomposites, we selected this coupling agent, although,  it has only two 2 polymerizable silanol groups.

B.  We agree  that  a di-silanol compound,   would be equivalent  to the well-known polymer chemistry result of a bifunctional monomer, which is only able to create linear polymers or loops,  but not a three dimensional network. In the present case, we have three different species involved, magnetite NPs with -OH, mesoporous silica with -SiOH, and hydrolized  APMDMS Si-(OH)2 where, hydroxil groups -OH can condensate creating covalent (Si-O-Si) siloxane bridges  that  provoke the grafting of magnetite NPs on the mesoporous silica surface (maybe with the creation of some intermediate  siloxane bridge loose loops).

Therefore, we agree with the Referee that the term layer, which one relates to the idea of a homogenous and appreciable network phase surrounding another material phase, was unfortunately used by us.

Accordingly, we have corrected in the text, the inexact term layer by grafting, which is a much more realistic description of the functionalization of magnetite NPS with the grafted  APMDMS, whose amine groups are involved in the coupling of  fluorescent Rhodamine molecules.
